# The Optical Coherence Tomography and Raman Spectroscopy for Sensing of the Bone Demineralization Process

**DOI:** 10.3390/s21196468

**Published:** 2021-09-28

**Authors:** Maciej J. Głowacki, Aleksandra M. Kamińska, Marcin Gnyba, Jerzy Pluciński, Marcin R. Strąkowski

**Affiliations:** Department of Metrology and Optoelectronics, Faculty of Electronics, Telecommunications and Informatics, Gdańsk University of Technology, 11/12 G. Narutowicza Str., 80-233 Gdańsk, Poland; maciej.glowacki@pg.edu.pl (M.J.G.); aleksandra_kaminska@o2.pl (A.M.K.); margnyba@pg.edu.pl (M.G.); jerpluci@pg.edu.pl (J.P.)

**Keywords:** bones, polarization-sensitive optical coherence tomography, Raman spectroscopy

## Abstract

The presented research was intended to seek new optical methods to investigate the demineralization process of bones. Optical examination of the bone condition could facilitate clinical trials and improve the safety of patients. The authors used a set of complementary methods: polarization-sensitive optical coherence tomography (PS-OCT) and Raman spectroscopy. Chicken bone samples were used in this research. To stimulate in laboratory conditions the process of demineralization and gradual removal of the hydroxyapatite, the test samples of bones were placed into 10% acetic acid. Measurements were carried out in two series. The first one took two weeks with data acquired every day. In the second series, the measurements were made during one day at an hourly interval (after 1, 2, 3, 5, 7, 10, and 24 h). The relation between the content of hydroxyapatite and images recorded using OCT was analyzed and discussed. Moreover, the polarization properties of the bones, including retardation angles of the bones, were evaluated. Raman measurement confirmed the disappearance of the hydroxyapatite and the speed of this process. This work presents the results of the preliminary study on the possibility of measuring changes in bone mineralization by means of the proposed methods and confirms their potential for practical use in the future.

## 1. Introduction

The human skeleton, throughout its lifetime, adjusts its size and internal structure to constantly changing biomechanical conditions. Due to their mechanical properties, bones protect soft organs and support the body, allowing it to control its position and to move. Additionally, the bones are an important reservoir of Ca^2+^ ions [1,2]. In the bone tissue, fixed processes of distribution and renewing of biochemical and structural components occur. Osteoblasts take part in formation and regeneration of the bones, while osteoclasts are involved in a resorption process. The macro-scale bone construction represents two forms: cancellous tissue and compact tissue. The latter creates an oriented structure composed of collagen fibers with a clear orientation.

The bone is a complex and not easily classifiable composite of organic substances and mineral compounds entangled on a nanostructured level [3,4,5]. The basic unit of the organic component of the bone is the type I collagen molecule, which is a long (300 nm) and thin (1.5 nm in diameter) protein composed of a triple helix. Two disparate ends can be distinguished in a polypeptide chain of the collagen. The end retaining a free amine group is called the N-terminus, while the end containing a free carboxyl group is called the C-terminus [6]. Repeated segments of five collagen molecules arranged parallel to each other in one plane at equal intervals of 0.24 nm, with every subsequent molecule in the segment being translated from the previous one by 67 nm, form a microfibril, which is presented in Figure 1 [4,7,8]. The N-termini of the succeeding 5-molecule segments are separated from the C-termini of the preceding ones by a distance (gap) of 36 nm [7]. A bunch of alternatively arranged microfibrils forms a fibril, which in turn is an element of a tissue fiber [7,9]. Since the neighboring microfibrils are aligned, the gaps in their structure create continuous channels [4,7].

The inorganic template of the bone is composed of hydroxyapatite, which predominantly occurs as a non-stoichiometric mineral of a hexagonal crystal system [10,11]. Hydroxyapatite is known to nucleate and grow within the gap channels, creating platelets of varying sizes and spatial orientations [3,7]. The crystals are uniaxially oriented along the long axis of the collagen molecules [3,4]. A significant amount of the mineral is also reported to exist in the extrafibrillar space [7]. Hydroxyapatite is a factor that introduces considerable irregularity to the otherwise hierarchical and highly organized structure of the collagen fibrils. The presence of both collagen and hydroxyapatite in the bones in an approximately 1:3 weight ratio makes the bones very strong and flexible at the same time.

Diverse types of bones bear the closest resemblance to each other on the nanostructured level [12]. Making the optical properties of the samples dependent on the changes in their nanostructure enables partial reduction of the impact of differences between specimens.

During ossification, two major changes, osteoclastogenesis and osteoblastogenesis, take place [13,14]. The balance between these processes is responsible for bone remodeling. During remodeling, the bone loss is closely coupled to the reconstruction process, which determines the proper functioning of the skeleton. Any imbalance between remodeling processes may cause serious diseases. Lower bone mineralization may be due to age (not all the bones are fully mineralized after birth), diseases (among which osteoporosis is best known), or injuries. Significant bone weakness leads to, for example, fractures or deformities. In some rare cases, excessive mineralization of the bones may occur, manifesting itself as a decrease in their elasticity. One of the main factors of the healthy state of the bone tissues is the proper management of calcium contained in hydroxyapatite.

In the clinical diagnosis, the healthiness of the bone is mainly examined using computed tomography (CT) with the Hounsfield method for tissue hardness evaluation [15,16,17]. In bone regeneration, the monitoring of tissue growth is a key element that determines the success of the treatment. Because of the high negative influence of X-ray radiation on the human body, the measurements are usually carried out with respect to the principal rule of ALARA (as low as reasonably achievable). For this reason, due to the need to limit the negative impact of ionizing radiation on living organisms during computed tomography examinations (5 to 100 times greater than conventional radiographs of the same area of the body), the bone tissue assessment cannot be performed more frequently than every 6 up to 9 months, which significantly extends the duration of treatment [17].

The purpose of this study is to determine whether bone demineralization can be estimated by means of complementary optical methods. As the bones are highly scattering media, the authors decided to use the optical coherence tomography (OCT, pioneered by David Huang, James G. Fujimoto, and co-workers in 1991 [18]). In addition to OCT, other measurement techniques for highly scattering media investigation, such as methods using integration spheres [19,20], time-of-flight spectroscopy [21,22], photoacoustic microscopy and spectroscopy [23,24], optical diffuse tomography [25,26], optical diffraction tomography [27,28], optical projection tomography (or optical transmission tomography) [29,30], and transmission optical coherence tomography [31,32], are known. However, these methods either may only be performed in vitro (which is a major limitation if future clinical applications are considered) or cannot provide additional information on bone properties (such as birefringence) with high spatial resolution, unlike polarization-sensitive OCT (PS-OCT).

OCT is an optical measurement technique used for investigations of a wide range of scattering materials [33]. OCT enables surface and subsurface examination of different types of materials to be performed in a non-contact and non-destructive way [34]. With the aid of OCT, one can analyze the depth structure of investigated materials with a measurement resolution of a few micrometers, high sensitivity, and a dynamic range [35]. Currently, the advantages of OCT make it applied in medical treatments, especially in ophthalmology, dermatology, stomatology, and endoscopy, and also in industry and science. The OCT measurements are based on selective, high-resolution detection of the light backscattered from the particular scattering points located inside the investigated object, which is needed for 2D and 3D tomography imaging. To perform such measurements, the OCT uses low-coherence interferometry (LCI). The LCI, also known as white light interferometry (WLI), is an attractive measurement method offering high measurement resolution, high sensitivity, and measurement speed. Since both hydroxyapatite and the collagen fibers are birefringent [36,37], it has been decided to use PS-OCT, which enables simultaneous measurements of the scattering properties of the bones and their polarization properties.

PS-OCT has already been used in investigations of hydroxyapatite-containing tissue specimens, namely teeth (tooth enamel is an anisotropic and weakly scattering medium that changes the state of polarization of backscattered optical radiation [12,38,39,40,41]), as well as collagen-containing tissues (e.g., skin [42,43], articular cartilage [44,45,46,47], meniscus [46], tendons [48], tumor [49,50], and atherosclerotic plaques [51] or coronary plaque [52] in blood vessels).

Raman spectroscopy has been used as a complementary method to verify the PS-OCT measurements. Raman spectroscopy is based on the illumination of the sample by a laser beam and spectral analysis of the inelastically scattered light. Interaction between the excitation light and oscillating molecular bonds results in the appearance of spectral lines which correspond to the distribution of oscillatory levels, which is unique to a particular material (e.g., hydroxyapatite). Thus, Raman spectroscopy provides crucial information about the molecular composition of investigated materials. The intensity of the Raman line is proportional to the concentration of oscillators (bonds) in an analyzed volume of the material, therefore reducing the content of the substance reduces the intensity of this line. Raman spectroscopy has been widely used for biomaterials investigation. For this purpose, infrared (IR) laser excitation is commonly used as this wavelength range enables significant reduction of fluorescence, which creates strong optical background interfering with the acquisition and processing of Raman signal. A significant advantage of the Raman measurement is the use of signals in the visible and near-infrared (VIS-NIR) spectral range. It enables the use of fiber-optic probes, which simplifies the coupling of the optical measurement system to the investigated object, as well as improves the safety of the measurements [53,54,55]. The more complex Raman techniques can be also used for biomaterials investigation. It includes surface-enhanced Raman spectroscopy (SERS) which enables a significant increase of usually weak Raman signal by use of additional interaction of laser radiation with selected particles [56]. Additionally, Raman microscopy with dedicated sample selection can be used to remove signals from unwanted sources [57]. However, these techniques require special sample pre-treatment, thus they were not introduced at this stage of the research as the aim of the research was to develop a minimally invasive sensing system.

Because the bones are highly scattering media (i.e., the bone scattering coefficient *μ*_s_ is much higher than the absorption coefficient *μ*_a_ [21,58,59]—in the near-infrared range, *μ*_s_ ≅ 6 mm^–1^ [21] and *μ*_a_ ≅ 0.02 mm^–1^ [21,58,59]; these values may vary between individuals), the use of PS-OCT or Raman spectroscopy systems for their examination, for example, for estimation of their demineralization, might be a difficult task. Moreover, as penetration depth of optical signals in the bones is limited, information is obtained mainly from area close to the surface. This is a preliminary study focused on finding the correlation between optical quantities of the bones, which can be measured with the aid of PS-OCT system, and the demineralization process (reduction of the hydroxyapatite) of the tested tissue. For ethical reasons, bones of animals previously intended for human consumption have been used. Conclusions from the conducted research, due to the similarity of bone structure (on a micro-scale), may be extended to human bones. Therefore, the presented research results can become an entry point for future research in the development of diagnostic methods assessing changes in the degree of bone demineralization caused by natural, disease, or injury factors during bone regeneration or fusion. These tests can also be the basis for the assessment of bone formation processes, e.g., around implanted hip or elbow joints, etc.

## 2. Experimental Procedure

### 2.1. Sample Preparation and Investigation Procedure

Extracted chicken humeri were used for the optical analysis of the demineralization process. Separate sets of the samples were prepared for measurements by PS-OCT and Raman spectroscopy. Diaphyses of the bones were thoroughly cleansed of muscle tissues while tendons and articular cartilage were removed from the epiphyses. To ensure repeatability of the scanned area, for every sample a thin transverse notch was made in the middle of the diaphysis. The bones were always measured from the notch toward the distal epiphysis. For PS-OCT examination the objects were additionally leveled. The appearance of the prepared humerus with a marked scanning area is presented in Figure 2.

Reference measurements were carried out with PS-OCT and Raman spectroscopy after the bones were rinsed with saline solution and dried in the open air. Subsequently, the samples were placed in 10% acetic acid to gradually remove hydroxyapatite from their structure. Two series of measurements were carried out, each on a separate set of the bones (two pieces—one for PS-OCT and one for Raman spectroscopy). The first group of the humeri was examined during the first 24 h of the demineralization, namely after 1 h, 2 h, 3 h, 5 h, 7 h, 10 h, and finally 24 h. The second set of samples was investigated every 24 h for 2 weeks using polarization-sensitive optical coherence tomography and Raman spectroscopy. In this study the samples were labeled as B24h_OCT (the sample used in the 24-h experiment for PS-OCT measurements), B24h_RS (the sample used in the 24-h experiment for Raman spectroscopy), B14D_OCT (the sample used in the 14-day experiment for PS-OCT measurements), and B14D_RS (the sample used in the 14-day experiment for Raman spectroscopy). The acetic acid greatly altered the mechanical properties of the samples. After 2 weeks of demineralization, the bones became flexible and could easily be bent.

### 2.2. PS-OCT and Raman Spectroscopic Measurement Systems

PS-OCT measurements were carried out with the use of a Polarization-Sensitive Swept-Source Optical Coherence Tomography (PS-SS-OCT) system with commercial laser swept source HSL-2000 (by Santec, Komaki, Japan). During each PS-OCT measurement, 128 horizontal sections (B-scans) were obtained for the single sample, with every image having been composed of 1024 axial scans (A-scans). The PS-OCT system delivers a depth-resolved map of scattering points inside the scanned sample with information on the intensity of backscattered light and the local changes of the retardation angle, which is related to the birefringent principles of the tested bone. Due to the bone demineralization—reduction of hydroxyapatite—the changes in the tissue optical scattering and birefringent features are expected. They should be visible in the PS-OCT images as well. To compare the measurement data at different stages of the experiment and to monitor the dynamics of the bone demineralization process, two parameters, the extinction coefficient (*µ*_e_) and the birefringence (Δ*n*), were used. They can be calculated based on the PS-OCT measurement data, which is explained in the next section (Section 3.2 and Section 3.3). The *µ*_e_ expresses the absorption and scattering features of the sample (*µ*_e_ = *µ*_a_ + *µ*_s_). The high value of *µ*_e_ for highly scattering materials gives the bright OCT images with low penetration in depth. Contrary, the materials with low *µ*_e_ return the lower backscattered optical signal, however, the depth of the sample penetration is higher, whereas, the Δ*n* refers to the optical birefringent features of the evaluated material and is linked to its inner structure at the microscopic level. The low Δ*n* is observed for random structures and optically isotropic media. Its high values are representative for oriented structures such as crystals or fibrils [37,60,61].

Raman spectroscopic system utilizes near-infrared (NIR) excitation wavelength to reduce the influence of fluorescence and other optical background signals. The Raman system is based on axial transmissive spectrograph and fiber-optic probe from the pre-commercial Raman system Ramstas developed by the VTT (Technical Research Center of Finland) [53,54]. The system also uses a diode laser coupled to an optical fiber. Power on the sample was limited to reduce the risk of its thermal damage. Collected scattered radiation was filtered to remove laser wavelength and transmitted to the spectrograph and thermoelectric-cooled CCD camera BV-401 (by Andor, Belfast, UK).

Detailed features of the measurement systems used in this study are presented in Table 1.

## 3. Measurements and Data Processing

### 3.1. Polarization-Sensitive Optical Coherence Tomography (PS-OCT) Measurements

All measurements of the evaluated samples of the chicken humeri were made with the aid of polarization-sensitive (PS) analysis applied to OCT. The PS analysis expands the OCT imaging by delivering information on the birefringence of the tested samples. These data are given in the form of the retardation angle, which can be understood as a phase shift between ordinarily and extraordinarily polarized optical beams in the birefringent medium. The PS-OCT system, which has been utilized in the experiment, estimates the retardation angle based on the Jones formalism, therefore, its value varies from 0 to π/2 (0 to 90 degrees). This implies the depth-resolved wrapped function of retardation for the birefringent medium [50]. An example of PS-OCT measurements of the chicken humerus is shown in Figure 3.

Standard OCT imaging usually expresses the changes of the backscattered light intensity as a grayscale cross-sectional image. Here, the retardation angle has been added, creating colorful multimodal OCT images. The intensity (value) of the image pixels corresponds to the level of backscattered light at a logarithmic scale, whereas the value of the retardation angle is coded by the color of the pixels (hue) according to the scale represented by the color bar.

During the experiment, 23 PS-OCT images were collected and processed for further analysis: 8 for sample B24h_OCT (taken in the first 24 h) and 15 for sample B14D_OCT (taken in the first 14 days). A representative set of the measurement results, which express the observed trends and support the conclusions, is shown in Figure 4.

The first attempt at the analysis was based on subjective evaluation of the PS-OCT cross-sectional images. Following, two main trends are observed. Firstly, exposure of the sample to acetic acid reduces the optical scattering and absorption effects inside the tested bone. This has been observed for both B24h_OCT and B14D_OCT samples as a decrease in the level of backscattered light from the bone’s inner structure in exposure time. To underline this conclusion, one should compare the images in Figure 4. The PS-OCT image, labeled as (a), was taken at the beginning of the experiment when the bone was not affected by the acid. Comparing this PS-OCT image with (b), (c), and (d), the lower intensity of backscattered light from the sample’s subsurface structure and higher penetration depth can be noticed. This phenomenon increases especially after a longer exposure time (longer than 1 or 2 days), which is confirmed by PS-OCT images (e) and (f) in Figure 4. Some valuable conclusions can be derived from the polarization-sensitive measurements. The first PS-OCT image in the series of B24h_OCT measurements expresses a random change in the retardation angle (the sample has not been influenced by acid yet), which suggests the light depolarizing nature of the evaluated structure. After 24 h of exposure of the bone to the acid, regular color stripes can be seen, which leads to the conclusion that the sample exhibited some birefringent effects. This and the scattering phenomena in the bones have been proceeded to quantitative analysis, described in the next session.

### 3.2. Analysis of the Backscattered Light Intensity

The intensity of the backscattered optical signal depends, among others, on the Fresnel reflections, and scattering and absorption coefficients of the evaluated tissue. All these factors can be expressed as a single parameter called the extinction coefficient. The relation between scattering features of the tissues and recorded OCT signals (A-scans) have been studied in [62] and [20,63]. When analyzing PS-OCT images that have been taken after long exposure of the tissue to acetic acid (see images (e) and (f) in Figure 4), a low level of backscattered light intensity (dark areas) between surface and subsurface interfaces can be observed. This phenomenon is a consequence of low tissue scattering and absorption in this particular case and leads us to expand the investigation by observing their changes in exposure time. Analyzing the scattering and absorption independently is difficult, especially using OCT methods. Therefore, the quantitative evaluation was based on the extinction coefficient, which represents both the effects.

The OCT recorded signal covers information about back-reflected and backscattered light at the surface and inside the evaluated sample. The first component refers to the reflection at the boundary between the sample and surrounding medium (e.g., the air), and between the layers composing the device under tests. This reflection is described by the Fresnel equations. The back-reflected signal occurs as rapid rise and fall of the value of the A-scan (single line of depth-resolved OCT scan), whereas the slower changes at the A-scan are related to the backscattered effects of the scanning light beam and are useful for the extinction coefficient estimation. The rate of the A-scan slope may indicate the regions at which the back-reflection or backscattering becomes a dominant effect. This has been expressed as an example presented in Figure 5.

Following the outcomes from [62], the OCT A-scan, as a function of depth, is described as the compound of an optical source point spread function (PSF) and Beer’s law. Due to the exponential character of backscattered light intensity, the extinction coefficient is calculated directly from the A-scan as a slope ratio of its natural logarithm.

The raw data from the OCT system are in the form of A-scans in the logarithmic scale (dB). The data processing started from defining the region of interests (ROI) in the OCT image, covering the B-scan from the marking notch toward the distal epiphysis, and limited at depth to the backscattered light intensity drop by 40 dB. The ROI useful for analysis was composed of about 850 A-scans, covering above 8 mm of lateral scanning distance from the edge of the notch. Afterward, all A-scans were adjusted to the point of light reflection from the surface of the evaluated sample, which clears out the tilt of the OCT image. Following, all A-scans inside the selected ROI were recalculated from the dB scale to the scale with a natural logarithm as a base, which corresponds to the mentioned Beer’s law. For every A-scan from the ROI, the extinction coefficient was estimated from the slope of the A-scan plot at the region where the light backscattering effects are dominant over the Fresnel reflection (Figure 5). Finally, the mean value of the extinction coefficient and its standard deviation were calculated. The above procedure has been applied to the measurement results obtained for samples B24h_OCT and B14D_OCT. The values of the averaged extinction coefficient as a function of time of exposure to the acetic acid are presented in Figure 6 and Figure 7.

The presented quantitative data express the dynamic change in the light scattering of the bones during the demineralization process, which is stimulated by exposure to the acid. After a few days of the test, the bone sample became almost transparent for the NIR radiation used in the experiment. The highest drop of the extinction coefficient was observed after the second hour of the acid influence on the bone. Comparing both plots together at the same points in time, there are slight differences in extinction coefficient values. At the beginning (0) it was estimated at 7.875 mm^–1^ for B24h_OCT (Figure 6) and 8.938 mm^–1^ for the B14D_OCT (Figure 7). After 24 h (one day) they were 1.878 mm^–1^ and 2.077 mm^–1^, respectively. The B24h_OCT and the B14D_OCT are the two different samples of the chicken humerus, therefore, this mismatch is related to the individual features of the tested bones. However, the course and dynamics of the observed process are similar for both samples.

### 3.3. Polarization-Sensitive Analysis in the Bones Evaluation

In the utilized PS-OCT system, the polarization-sensitive measurement data are delivered in the form of depth-resolved wrapped function of the retardation angle with a periodicity of π. The retardation angle exposes the phase shift between linear orthogonal components of polarized light guided through a birefringent medium. Due to the birefringent nature of collagen and hydroxyapatite, this effect was expected in our observation. In this particular case, birefringence (Δ*n*), defined as the difference between refractive indexes for both polarization orthogonal components (known as ordinary and extraordinary rays), is the best approach to express the polarization effects in the bones. The Δ*n* can be calculated easily from the derivative of the retardation angle function. This is shown in Figure 8.

The birefringence Δ*n* also results from the retardation provided by the tested sample at the specific geometrical path *z*, which is expressed as:(1)Δn=Γz
or
(2)$$\Delta n=(\gamma \cdot \lambda_{0})/(2\pi \cdot z),$$
where: *γ* is the retardation angle, Γ is the retardation (expressed in length units) as a delay between polarization orthogonal components at a specific geometrical path *z*, and $\lambda_{0}$ is the central wavelength of the scanning beam (1320 nm for the utilized PS-OCT system).

Depending on the plot of the retardation angle, two different approaches were applied. If the two neighboring maxima or minima were recognized, the Δ*n* was calculated based on Equation (1), where provided retardation was equal to half of the central wavelength and *z* was the distance between these maxima. Otherwise, the birefringence estimation was performed by analyzing the slope of the plot, which is presented in Figure 8 as a dashed line. In this case, the Δ*n* was calculated as:(3)$$\Delta n\;=\;(\Delta \gamma \cdot \lambda_{0})/(2\pi \cdot \Delta z)$$
where Δ*γ* is the retardation angle change over the path length Δ*z*.

The raw data were in the form of a two-dimensional map of the retardation angle, which covers the cross-sectional OCT image inside the ROI selected for the “Analysis of the backscattered light intensity”, described in the previous subsection. The data processing starts from eliminating bright and dark spots (locally extremely high and low values) using a median filter. As previously, each line (A-scan) of the retardation map was adjusted according to the surface of the tested bone and filtered by a low-pass Gaussian filter to obtain a smooth depth-resolved function of the retardation angle. For every line, the birefringence as the Δ*n* was calculated based on the procedure described above. Afterward, the average value of Δ*n* and its standard deviation were calculated. The change in birefringence as a function of time of exposure to acid is presented in Figure 9 and Figure 10.

At the beginning of the experiment, both tested samples (B24h_OCT and B14D_OCT) presented a high ability to depolarize the scanning light beam, which occurred as a random change of the value of the retardation angle observed at PS-OCT B-scans (Figure 4a) and the value of calculated birefringence Δ*n* being close to 0. After a short time of exposure to the acid (about 1 h), the bones exhibited some birefringent-like behavior, which is observed as a rising value of Δ*n*. For the next hours, the Δ*n* was rising rapidly. This process took about two days from the beginning of the experiment. Afterward, the fall of birefringence was observed and after about eight days its value became stable.

After one day of the experiment (24 h) both tested samples exhibited different values of the birefringence. This mismatch is related to the individual optical features of the bones, which has also been discussed in the previous Section 3.2.

### 3.4. Raman Spectroscopy

Raman spectra have been processed with the use of OriginLab OriginPro software. Selected Raman spectra recorded during the experiment taking 24 h (sample B24h_RS) and 14 days (sample B14D_RS) are shown in Figure 11. The main Raman band at 967 cm^–1^ has already been reported in the literature [64,65]. It is assigned to *ν*(PO_3_) symmetric stretching of hydroxyapatite. As the strongest band assigned to hydroxyapatite, it was used as a marker for quantitative estimation of demineralization of the bone resulting from its interaction with the acetic acid.

The quality of the raw data obtained from Raman measurements is lowered by optical background originating mainly from fluorescence, which is common in the investigation of biological materials. Before the quantitative analysis of the recorded signal, this component was removed from the Raman spectra by polynomial approximation and subtraction. For quantitative analysis, the integral intensity (area) of the band was calculated with the use of the Lorenz function. The result of the integral intensity calculation was divided by the intensity value for a different Raman shift value that should have remained constant during the experiment (760 cm^–1^ was selected as an example) to overcome the small defocusing effect caused by the complex surface of the sample. Then, the results were normalized in relation to the values determined for initial measurements in order to quantify the disappearance of hydroxyapatite in the investigated area of the bone. Results are shown in Figure 12.

The time courses of normalized intensity changes of the Raman bands at 967 cm^–1^ are in good agreement with the results of the OCT investigation. As a result of the integration of the bone with the acid the intensity of the line assigned to hydroxyapatite quickly decreases. This indicates the demineralization of the sample, caused by the removal of the hydroxyapatite. Despite the small differences in the speed of this process between two samples (the sample evaluated during the 24 h and the second one measured for 14 days), caused by individual variation, one can observe that it was the most intense during the first day (decrease to about 20–40% of initial values), especially during the first three hours. After about 5–7 days the Raman signal assigned to the hydroxyapatite cannot be noticed. The character of changes was additionally approximated by the exponential function. There is a good time correlation between the decrease of the extinction coefficient of the bones (see Figure 6 and Figure 7) and the removal of the hydroxyapatite (see Figure 12).

## 4. Discussion

Following the analysis of the PS-OCT images, the optical scanning depth is about 1 mm, which limits the investigation to the periosteum and a part of compact bone. The reliability of obtained results was also verified by the research work of Nadya Ugryumova et al. from 2004 [20]. In the quoted paper the values of the extinction coefficient of healthy bone, estimated from the OCT measurements, are in good agreement with those presented in this work. In [20] (Figure 12) the average value of µ_e_ is about 6 mm^–1^, while estimated by us are in the range of 7 to 9 mm^–1^.

All PS-OCT and Raman observations were performed with reference to the sample evaluated at the beginning of the experiment (Figure 4a). For this sample, the measurements were taken before it had been influenced by the acetic acid. In this case, the obtained PS-OCT result expresses a typical tomographic image for highly-scattering, isotropic medium: (i) lower penetration of the scanning beam in depth compared to the other PS-OCT measurement results, (ii) high backscattered signal from subsurface layers, and (iii) retardation angle changing randomly. In comparison with the reference data, the other PS-OCT images exhibit some birefringent features of the sample with the reduced volume of hydroxyapatite. Moreover, the sample becomes more transparent in the NIR radiation range, which has been shown by observing the distinctly lower intensity of backscattered light.

These conclusions are confirmed by the analysis of the extinction coefficient (Figure 6 and Figure 7) and the change in birefringence (Figure 9 and Figure 10) of the evaluated sample in the first 24 h and for the longer period of 14 days. For the evaluation of sample scattering abilities, the extinction coefficient *μ*_e_ becomes a key factor. Referring to the experiment, the highest *μ*_e_ was noticed for the first measurements at the beginning, where the evaluated sample highly scattered the scanning beam. Afterward, the *μ*_e_ was decreasing, which might be explained by the optical clearance of the sample by acetic acid infiltration. Due to its optical high-scattering features [66], the healthy bone exhibits high values of the extinction coefficient. A noticeable decrease in the *μ*_e_ caused by lower scattering was observed, which correlates with OCT images and Raman spectroscopic measurements, and is also supported by conclusions in [20]. The rapid fall of the *μ*_e_ values occurred in the first hour. After 3 h the process was stabilized and its dynamics became much lower than in the first 3 h. After seven days the bones became almost completely transparent for the scanning optical beam. The reason for the decrease in scattering coefficient *μ*_s_, which results in a reduced extinction coefficient *μ*_e_, is that acetic acid leaches hydroxyapatite from the bone (hydroxyapatite has a much higher scattering coefficient than collagen, which acetic acid does not leach). The leaching rate of hydroxyapatite is the greatest in the first few hours when hydroxyapatite is removed from the areas close to the surface of the bone. The leaching of hydroxyapatite from deeper areas of the bone takes much longer, which causes a slowdown in reducing the scattering coefficient *μ*_s_ (and the extinction coefficient *μ*_e_)—as confirmed by the measurements carried out.

The composition of hydroxyapatite and the collagen fibrils makes the bone very durable, as well as highly scattering and depolarizing. However, both components have birefringent features, therefore, monitoring of the birefringence (Δ*n*) has been carried out. At the beginning (the zero time) the value of Δ*n* was low. It means that the sample did not exhibit birefringence features or was highly scattering, which depolarized the scanning beam. After putting the sample under the acetic acid influence the significant growth of the birefringence was observed. Referring to Figure 9 and Figure 10, the phenomenon was observed for the first 3 h of the hydroxyapatite reduction process and continued for the next days. After the second day of the experiment, the birefringence reached its maximum and slowly fell to reach its stable value after the eighth day of the experiment.

The birefringence (Δ*n*) and the extinction coefficient (*μ*_e_) are sensitive to the changes in the bone structure and its biochemical composition. With a gradual removal of the inorganic matrix (hydroxyapatite), monitored with the aid of Raman spectroscopy, the *μ*_e_ decreased, while Δ*n* designated significant growth.

The comparison of the results recorded with the OCT method and Raman spectroscopy, which are characterized by good timing, indicates the possibility of explaining changes in the optical parameters of the bones through changes in their internal structure—removal of the hydroxyapatite. The process during the first days was very intense especially in the layer of the near-surface bone. Removal of the hydroxyapatite skeleton, which due to its orientation (see Figure 1) disturbs bone birefringence, causes the material to become more birefringent due to the parallel orientation of the collagen fibers. This process is accompanied by an increase in the penetration depth of the measurement signals and birefringence changes observed by the PS-OCT method and some Raman signal fluctuations, which result from the fact that the measurement signals began to contain a composition of deeper layers, in which residual hydroxyapatite still remained.

After about two weeks, the sample became flexible, which additionally confirms the hypothesis about the demineralization processes taking place, and at the same time shows that after this time they covered the entire volume of the examined bone. Since the process of removing hydroxyapatite involves complex bone structure, it is necessary also to take into account the possibility of intermediate states associated with successive demineralization and change in bone structure, which may result in some shifts between the characteristics of changes in the Raman signal and PS-OCT as a function of time. This requires further research related to the precise calibration of the sensor technique but does not undermine a verified relationship between the removal of hydroxyapatite and changes in bone optical parameters that can be seen in the OCT images.

This confirms the possibility of using bimodal PS-OCT and Raman measurements for early determination of changes associated with demineralization of the bones or comparative study of their mineralization level. However, it is too early to predict the usefulness of this method in clinical diagnosis. Here, the correlation between bones mineralization level and the optical parameters measured by PS-OCT has been shown, but the limits of the PS-OCT, such as low scanning depth (in this case 1 mm for the healthy bone) and a need for direct access to the evaluated tissue may cause a serious problem for in-vivo diagnosis. This study should be treated as preliminary research, where the problems such as the interface to the bones in living organisms or low scanning beam penetration depth are not the issue, making an entry point for future research.

## 5. Conclusions

This study presents the possibilities to assess the degree of hard tissue demineralization. The use of the dedicated PS-OCT and Raman system equipped with IR sources for the investigation of the bones enables the acquisition of sufficient intensity signals and reduced impact of optical interferences. According to obtained results, the strong correlation between the level of the hydroxyapatite reduction in the bones monitored with the aid of Raman spectroscopy and the polarization-sensitive optical coherence tomography imaging can be observed. Likewise, the dynamics of the bone demineralization process can be measured successfully by both PS-OCT and Raman spectroscopy. This PS-OCT method may have application potential in the monitoring of resorption and regeneration process of the bones as well as early detection of pathological changes. It can be used during bone recuperation after, e.g., bone injury, surgery, or implant application. This report presents the preliminary results, which discover the capabilities of the PS-OCT for bone diagnosis. More research work is needed for quantitative data recovery from PS-OCT imaging to make this technique useful in clinical practice. Moreover, the problem of the bones imaging through skin and muscles using PS-OCT has not been solved yet. All measurements presented in this report have been performed ex vivo. For now, direct access to the surface of the bone is required, which becomes a key problem to overcome. Obviously, this method can be used during surgery, where easy access to the bone is maintained. Furthermore, there are some studies in which the admittance to the evaluated hard tissue is provided by the optical interface through the implant [67]. However, further studies on optical interfaces to the bones, as well as better optical signal acquisition and recovery are needed. Moreover, for the wide application of the optical method in the investigation of tissues and other biological materials, it is necessary to compare and optimize the performance of different PS-OCT and Raman measurement systems [68,69,70,71]. In the further stage, more precise quantitate calibration of developed photonic sensing system will be also carried out with the use of well-established methods such as densitometry or X-ray investigation. Moreover, correlation between demineralization processes which take part close to the bone surface and those that take part in whole volume of the bones will be studied in detail.

## Figures and Tables

**Figure 1 sensors-21-06468-f001:**
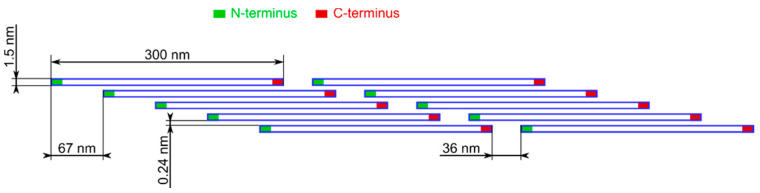
Arrangement of collagen in the microfibril. The diameters of the proteins and the 0.24 nm spaces between the molecules are disproportionately large for the illustration.

**Figure 2 sensors-21-06468-f002:**
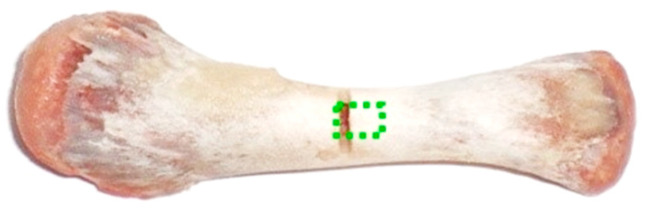
Extracted chicken humerus prepared for the reference OCT measurement. The green rectangle indicates the scanned region.

**Figure 3 sensors-21-06468-f003:**
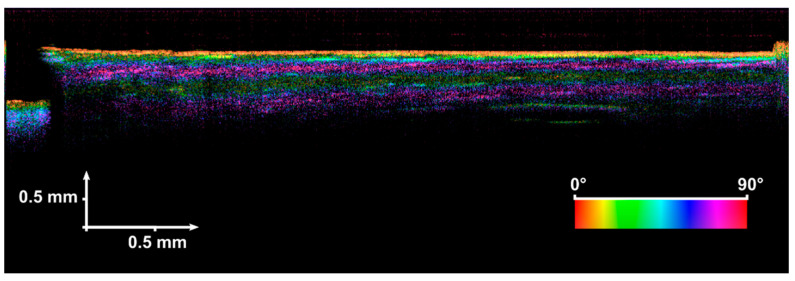
An example of PS-OCT imaging of the chicken humerus after 1 day of being exposed to 10% acetic acid (sample B14D_OCT).

**Figure 4 sensors-21-06468-f004:**
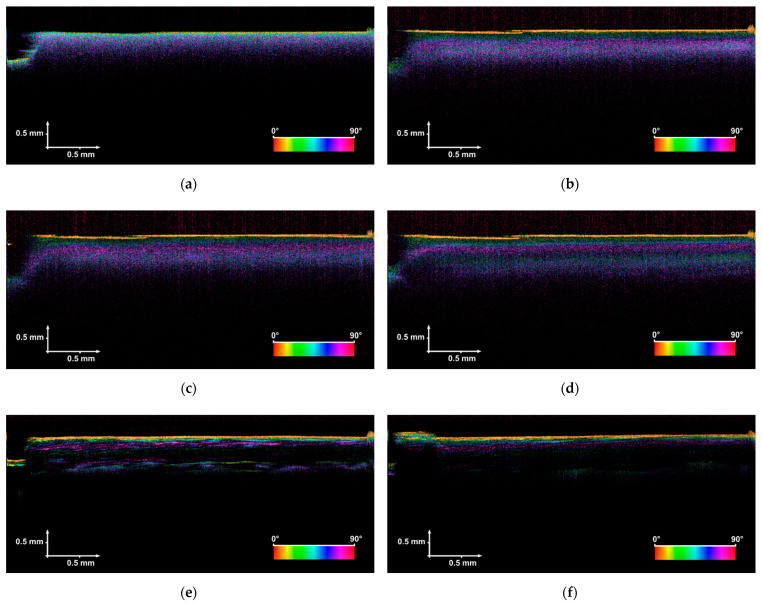
PS-OCT images of the tested bones taken in the first 24 h (sample B24h_OCT, figures (**a**–**d**)) and 14 days (sample B14D_OCT, figures (**e**,**f**)); (**a**) the image taken at the beginning of the experiment (the bone has not been exposed to the acetic acid), (**b**) after 1 h; (**c**) after 3 h; (**d**) after 24 h, (**e**) after 7 days, (**f**) after 14 days.

**Figure 5 sensors-21-06468-f005:**
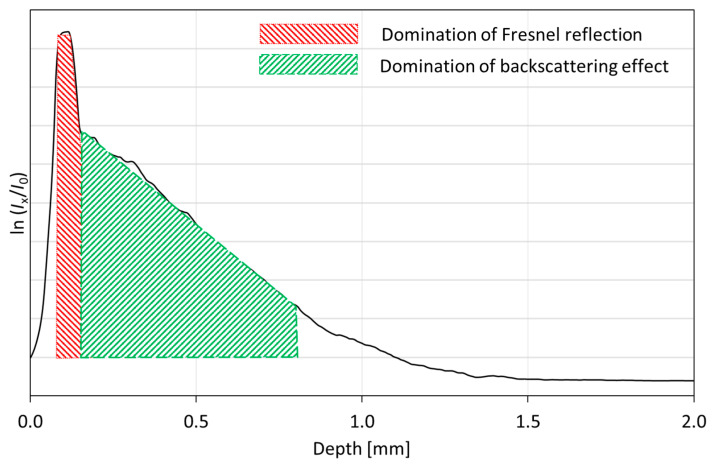
An example of OCT A-scan, at which the regions of Fresnel reflection and backscattering domination are indicated; $I_{x}$ —the intensity of backscattered/back-reflected optical radiation, $I_{0}$ —reference intensity, usually the intensity of the incoming light beam.

**Figure 6 sensors-21-06468-f006:**
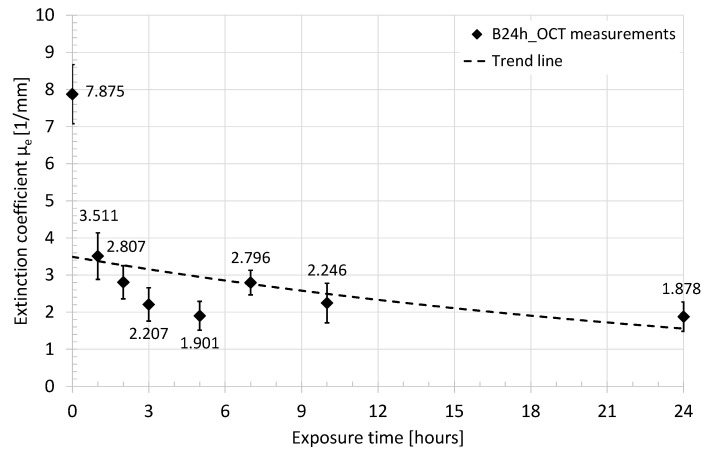
The average value of the extinction coefficient measured in the first 24 h of the experiment—sample B24h_OCT.

**Figure 7 sensors-21-06468-f007:**
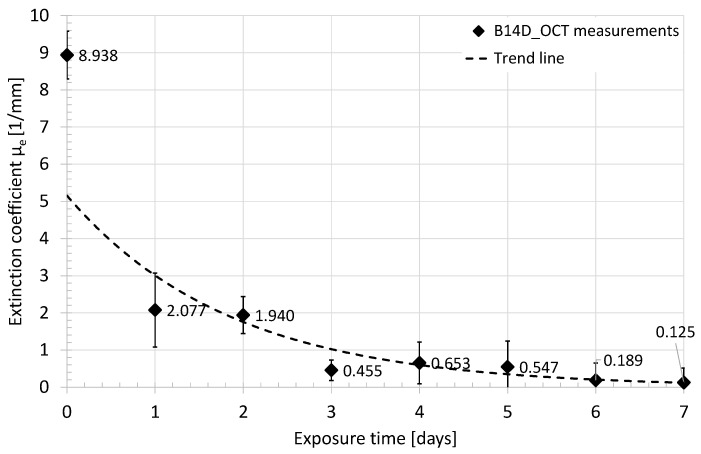
The average value of the extinction coefficient measured in the 14-day period—sample B14D_OCT.

**Figure 8 sensors-21-06468-f008:**
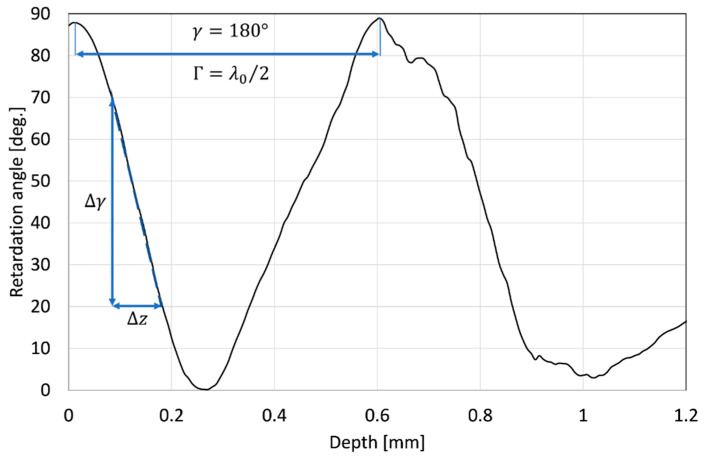
An example of a plot of the retardation angle taken for a single A-scan.

**Figure 9 sensors-21-06468-f009:**
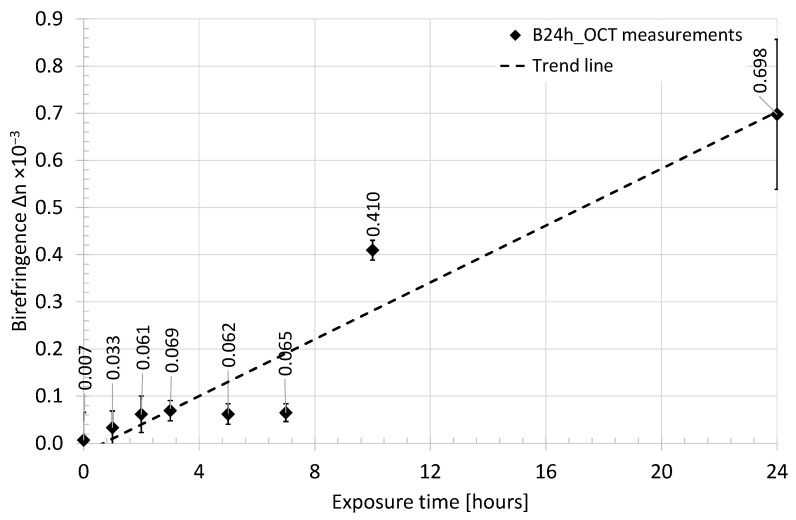
The average value of birefringence of the chicken humerus recorded in the first 24 h of exposure to acid—sample B24h_OCT.

**Figure 10 sensors-21-06468-f010:**
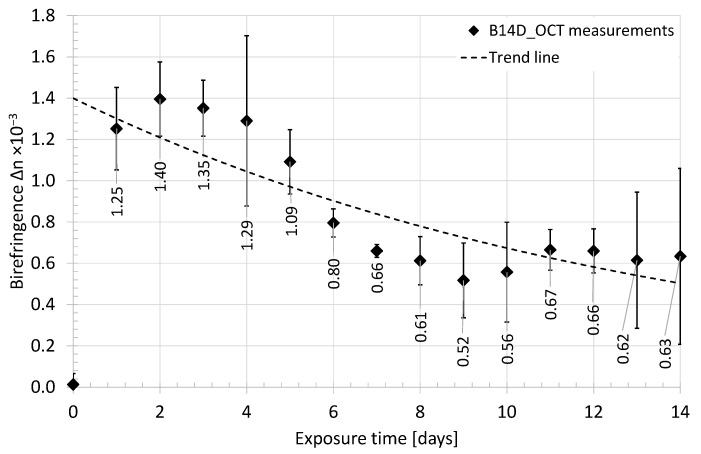
The average value of birefringence of the chicken humerus recorded for 14 days of exposure to acid—sample B14D_OCT.

**Figure 11 sensors-21-06468-f011:**
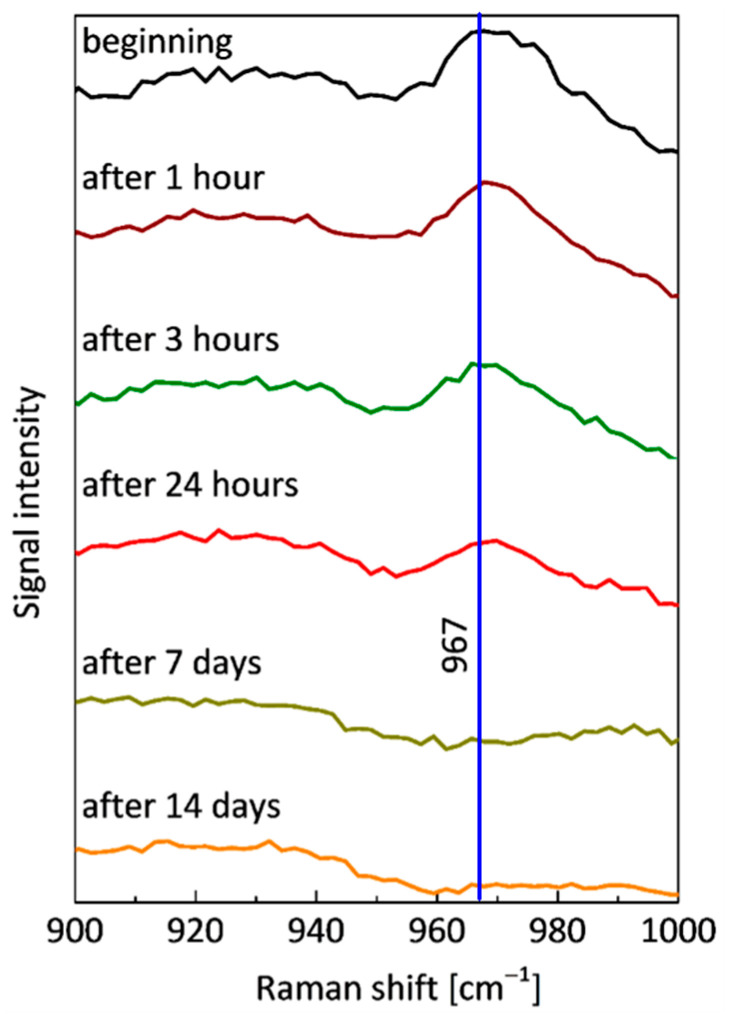
Selected Raman spectra of the tested bone taken for the first 24 h and 14 days—the spectrum taken at the beginning of the experiment (the bone has not been put at the acetic acid influence) and subsequently after 1 h, after 3 h, after 24 h, after 7 days and after 14 days.

**Figure 12 sensors-21-06468-f012:**
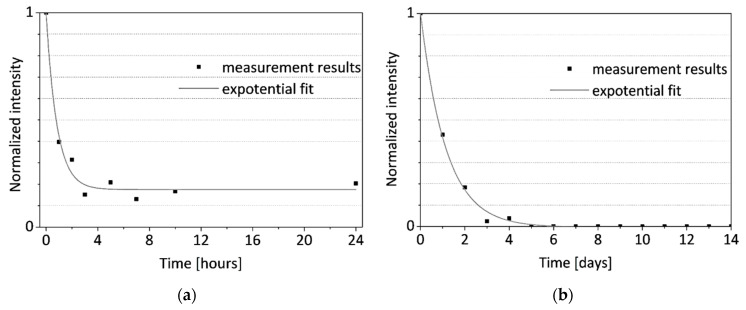
The time courses of normalized intensity changes of the Raman band at 967 cm^–1^ during (**a**) sample B24h_RS 24-h experiment and (**b**) sample B14D_RS 14-day experiment.

**Table 1 sensors-21-06468-t001:** Features of the PS-OCT system and the Raman spectrometer.

PS-OCT System		Raman Spectrometer	
Item	Value	Item	Value
Light source type	Diode laser—20 kHz swept source	Light source type	Diode laser—CW mode
Average output power	10 mW	Average output power	100 mW
Central wavelength	1320 nm	Central wavelength	830 nm
Wavelength range	140 nm	Raman spectral range	Stokes band—200–2000 cm^−1^
Longitudinal resolution	12 µm	Spectral resolution	8 cm^−1^
Lateral resolution	15 µm	Spectrograph	Axial transmissive setup with a holographic transmission grating
Frame rate	>4 fps	Detector	thermoelectric-cooled CCD array —1024 columns, −50 °C
Depth imaging range	7 mm	Optical system	Fiber-optics probe—a working distance of 5 cm
Transverse imaging range	10 mm		

## Data Availability

The data presented in this study are available on request from the corresponding author. The data are not publicly available due to privacy issues.

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
