# Peer review of "The Optical Coherence Tomography and Raman Spectroscopy for Sensing of the Bone Demineralization Process"

_sensors, 2021, doi:10.3390/s21196468_

Round 1
Reviewer 1 Report
1, In this manuscript, the authors used both PS-OCT and Raman spectroscopy to analyze the bone demineralization process. Although both imaging methods showed similar trends, it is still lack of gold standard to evaluate the accuracy of these two methods. I would suggest the author add another experiment measure bone mineral density (BMD) with gold standard, which will also help to enhance the significance of this manuscript.
2, Regarding the data analysis part, I think the authors should have done several measurements for each sample with different locations at different time points. According to the analysis results, such as figure 6-7, 9-10 and 12, the authors only shown average value or single value. It would be better to include error bar with these figures, which will make the results look more convincing. In addition, the authors need to clarify how many samples were included in this study. If the number is larger than 1, a statistic result needs to be included.
3, The figure 7 is the same as figure 6. The authors may misplace the figure 7.
4, As shown in the results, the penetration depth for PS-OCT is around 500um. Since it can only detect the surface of the bone, will it be a big problem to the inner structure of the bone. The author should include it in discussion.
5, The 24hours result with sample B24h_RS is not consistent with the 1-day result of sample D24h-RS. It also happens with the sample B24h_OCT and D24h_OCT. The authors need to clarify this. It looks like this is due to the low sample number, if so, the authors need to increase the sample number and add statistical results.
Author Response
Thank you for your valuable comments. We have attached our answers below. We made changes to the manuscript based on them.
1, In this manuscript, the authors used both PS-OCT and Raman spectroscopy to analyze the bone demineralization process. Although both imaging methods showed similar trends, it is still lack of gold standard to evaluate the accuracy of these two methods. I would suggest the author add another experiment measure bone mineral density (BMD) with gold standard, which will also help to enhance the significance of this manuscript.
Answer & Actions:
We agree with the reservations regarding our research methodology, but in the article we present the results of preliminary work which are to demonstrate the ability of the applied optical methods to register useful signals and provide information on bone mineralization. Our methodology at this stage of the research relied on the use of Raman spectroscopy and manual bone softening observation as reference methods for OCT. Work will continue for more samples and with the introduction of further reference methods to enhance biostatistics.
Reference methods and the plans for their use in further works were mentioned in the introduction to the article and the summary of the work.
2, Regarding the data analysis part, I think the authors should have done several measurements for each sample with different locations at different time points. According to the analysis results, such as figure 6-7, 9-10 and 12, the authors only shown average value or single value. It would be better to include error bar with these figures, which will make the results look more convincing. In addition, the authors need to clarify how many samples were included in this study. If the number is larger than 1, a statistic result needs to be included.
Answer & Actions:
We agree with this opinion. The PS-OCT images are composed of series of A-scans. For each A-scan the quantities, birefringence and extinction coefficient, were calculated. The average values and standard deviation were added and presented in the figures 6, 7, 9, 10, which fulfilled the requirements for statistical analysis. This has been explained in section 3 “Measurements and data processing”.
Following the Reviewer suggestions, the OCT image (B-scan) covers about 8 mm scanning line in the lateral (along the shaft) which provide the recommendation to measure in different points.
3, The figure 7 is the same as figure 6. The authors may misplace the figure 7.
Answer & Actions:
Thank you. It is an obvious mistake. Correct figures have been added to the manuscript.
4, As shown in the results, the penetration depth for PS-OCT is around 500um. Since it can only detect the surface of the bone, will it be a big problem to the inner structure of the bone. The author should include it in discussion.
Answer & Actions:
We agree with this comment. Low depth of penetration is a problem of the optical interface to the bone. For this reason, the results mainly relate to layers close to the surface, which limits the range of applications, but does not eliminate it (e.g. the possibility of examining open fractures). In the next phase of work, we plan to investigate the in-vitro correlation between the processes taking place at the bone surface and in its deeper layers, as mentioned in the summary.
5, The 24hours result with sample B24h_RS is not consistent with the 1-day result of sample D24h-RS. It also happens with the sample B24h_OCT and D24h_OCT. The authors need to clarify this. It looks like this is due to the low sample number, if so, the authors need to increase the sample number and add statistical results.
Answer & Actions:
We agree. The mismatch between OCT results after 24h and 1d was clarified in 3 (the end of 3.2 and 3.3) and it is related to the individual features of the bones. This is a consequence of the adopted methodology, but it points to the need to take into account the individual variability of the samples and to extend the biostatistics, which is planned in further stages.

Reviewer 2 Report
In the sensors- 1320892 manuscript, the authors report on a study by polarization-sensitive Optical Coherence Tomography and Raman spectroscopy for sensing of the bone demineralization process. To stimulate the process of demineralization and gradual removal of the hydroxyapatite, the test samples of chicken bones were placed into 10% acetic acid. Raman spectroscopy was used to evaluate the disappearance of the hydroxyapatite. However, probably a conventional technique for evaluating the effect of acetic acid treatment would have been useful. The relation between the content of hydroxyapatite and images recorded using OCT was analyzed and discussed. According to the authors, the polarization properties of the bones were evaluated.
In the present form, the paper is not clear and cannot be deeply evaluated, also because it should be rewritten in a clearer form. The scientific aim of the work is not clear.
The abstract has to be deeply revised. The introduction is very confused. some statements are not well-supported and/or not informative (i.e., “Bones in human organisms play several important roles. They form a skeleton – a living structure, capable of growth, adaptation, and repair”; in the first paragraph there is no reference; “The C termini of the succeeding segments are separated from the N termini of the preceding ones by a distance (gap) of 36 nm”). The introduction does not help the reader understanding the aim of the work. Given the statement “Because the bones are highly scattering media, the use of PS-OCT or Raman spectroscopy systems for their examination, for example, for estimation of their demineralization, could have been a much more difficult task. “, what is the aim of the paper?
The introduction is not exahustive (for instance, the authors do not report that there are various techniques for studying scattering media: OCT is not the only one).
Many details on experimental set-up are missed (including not explained acronym used).
There are obscure statements (see, for instance, “For the continuous increase of the retardation value with depth, the retardation angle becomes a periodic function, which can be seen as regular color stripes in Figure 3.”)
In section 3, the authors should introduce the meaning of the estimated parameters to help the reader in understanding the meaning of their results. For instance, the retardaton angles are not well introduced, despite they seem to be crucial. the Jones formalismi s cited but no explanation and/or reference is given. In this section is said that “Following the experimental procedure, described in section 2, ……” but no description is reported in section 2.
The discussion of the results is not convincing. The authors say “The first attempt at the analysis was based on subjective evaluation of the PS-OCT cross-sectional images. Following, the two main trends are seen. Firstly, sample exposure to acetic acid reduces the optical scattering and absorption effects inside the tested bone.” But it is not clear how the authors evidence this behavior.
A clear description of the data analys procedure is strictly necessary.
The quality of reported Raman spectra is very poor.
What the authors mean for optical offest?
The procedure described at page 12, lines 1-7 below Fig.11 is not clear.
Other comments
The rationale of Table 1 is not clear.
How the authors evaluate the Extinction coefficient?
What is the meaning of the statement “The time courses of normalized intensity changes of the Raman bands at 967 cm–1 are in good agreement with the results of the OCT investigation.”?
In Fig. 12 an exponential (even if it is called “expotential£) fit is reported. The datasets seem to be too small for such a kind of fit.
The authors say that “The comparison of the results recorded with the OCT method and Raman spectros-copy, which are characterized by good timing, indicates the possibility of explaining changes in the optical parameters of the bones through changes in its internal structure – removal of the hydroxyapatite.”. This is not supported by data reported and related discussion.
Author Response
Thank you for your valuable comments. We have attached our answers below. We made changes to the manuscript based on them.
- In the sensors- 1320892 manuscript, the authors report on a study by polarization-sensitive Optical Coherence Tomography and Raman spectroscopy for sensing of the bone demineralization process. To stimulate the process of demineralization and gradual removal of the hydroxyapatite, the test samples of chicken bones were placed into 10% acetic acid. Raman spectroscopy was used to evaluate the disappearance of the hydroxyapatite. However, probably a conventional technique for evaluating the effect of acetic acid treatment would have been useful. The relation between the content of hydroxyapatite and images recorded using OCT was analyzed and discussed. According to the authors, the polarization properties of the bones were evaluated. In the present form, the paper is not clear and cannot be deeply evaluated, also because it should be rewritten in a clearer form. The scientific aim of the work is not clear.
Answer & Actions:
We gratefully acknowledge the Reviewer for these remarks. We added additional description of the aim of the research to the main body of the manuscript.
- The abstract has to be deeply revised.
Answer & Actions:
We gratefully acknowledge the Reviewer for these remarks. We rewrote the abstract to make it more clear for the Readers.
- The introduction is very confused. some statements are not well-supported and/or not informative (i.e., “Bones in human organisms play several important roles. They form a skeleton – a living structure, capable of growth, adaptation, and repair”; in the first paragraph there is no reference; “The C termini of the succeeding segments are separated from the N termini of the preceding ones by a distance (gap) of 36 nm”).
Answer & Actions:
We gratefully acknowledge the Reviewer for these remarks. The first paragraph of the “Introduction” chapter has been revised in order not to include absolutely obvious facts and examples, like the sentence quoted by the Reviewer, regarding the fact that bones form the skeleton. The first paragraph does not contain references, as the Reviewer rightly pointed out. This is because the paragraph briefly summarizes important functions of the bones, which are not recent scientific discoveries but rather a common knowledge available in curricula. Since the manuscript already contains 40 references, we were cautious not to unnecessarily over-expand the bibliography.
In order to make the spatial structure of collagen molecules in the bones easily understood for the Readers, the second paragraph of the “Introduction” chapter has been expanded. Definitions of the N-terminus, and the C-terminus in a polypeptide chain of the collagen have been introduced and supported by a new reference:
Holmes, R., Kirk, S., Tronci, G., Yang, X., Wood, D. Influence of telopeptides on the structural and physical properties of polymeric and monomeric acid-soluble type I collagen. Materials Science and Engineering C 2017, 77, 823–827.
https://doi.org/10.1016/j.msec.2017.03.267
Moreover, Figure 1 has been updated to show the exact locations of the C-termini and the N-termini, to unambiguously illustrate the sentence quoted by the Reviewer, regarding the 36-nanometer gap between the molecules.
- The introduction does not help the reader understanding the aim of the work.
Given the statement “Because the bones are highly scattering media, the use of PS-OCT or Raman spectroscopy systems for their examination, for example, for estimation of their demineralization, could have been a much more difficult task. “, what is the aim of the paper?
The introduction is not exahustive (for instance, the authors do not report that there are various techniques for studying scattering media: OCT is not the only one).
Many details on experimental set-up are missed (including not explained acronym used).
There are obscure statements (see, for instance, “For the continuous increase of the retardation value with depth, the retardation angle becomes a periodic function, which can be seen as regular color stripes in Figure 3.”)
It was rewritten in a more clear form in subsection 3.1
In section 3, the authors should introduce the meaning of the estimated parameters to help the reader in understanding the meaning of their results. For instance, the retardation angles are not well introduced, despite they seem to be crucial. the Jones formalism is cited but no explanation and/or reference is given. In this section is said that “Following the experimental procedure, described in section 2, ……” but no description is reported in section 2.
The meaning of the estimated parameters based on PS-OCT measurements have been explained in section 2: (2.2). The problem of retardation angle has been rewritten in the clear form with giving the reference (the end of the first paragraph in subsection 3.1).
The discussion of the results is not convincing. The authors say “The first attempt at the analysis was based on subjective evaluation of the PS-OCT cross-sectional images. Following, the two main trends are seen. Firstly, sample exposure to acetic acid reduces the optical scattering and absorption effects inside the tested bone.” But it is not clear how the authors evidence this behavior.
A clear description of the data analys procedure is strictly necessary.
Answer & Actions:
We gratefully acknowledge the Reviewer for these remarks. We rewrote the related chapters to give more details about PS-OCT and make this part more clear for the Readers.
The quality of reported Raman spectra is very poor.
What the authors mean for optical offest?
The procedure described at page 12, lines 1-7 below Fig.11 is not clear.
Answer & Actions:
We gratefully acknowledge the Reviewer for these remarks. Additional explanations were given in the text to explain problems connected with recording and processing of Raman spectra of biomaterials, which usually contains unwanted components”. It starts from “The quality of the raw data obtained from Raman measurements is lowered by optical background originating mainly from fluorescence, which is common in the investigation of biological materials. Before the quantitative analysis, of the recorded signal, this component was removed from the Raman spectra by polynomial approximation and subtraction.” We removed optical background, but as such approach usually gives some remnants of this background (fluorescence, etaloning etc.) line 760 cm–1 was selected as an example of position in spectrum without Raman components and intensity of line at 967 cm-1 was related to intensity of this position to avoid influence of fluctuations of optical background caused e.g. by changes of depth of penetration of optical signals.
Other comments
The rationale of Table 1 is not clear.
Answer & Actions:
We are happy to explain the rationale of Table 1. Taking into consideration a comfort of the Readers who would like to reproduce or expand the experiment described in the article, we have decided to present essential parameters of the measurement systems in a form of table for two reasons. Firstly, this manner makes the information very easy to localize in the manuscript. Secondly, the parameters listed in the table can be easily compared with corresponding parameters of other systems, or with specification sheets.
How the authors evaluate the Extinction coefficient?
Answer & Actions:
The estimated values of extinction coefficient have been compared with the results published in the quoted reference. Sufficient explanations have been introduced at the beginning of the Discussion section: “The reliability of obtained results were also verified by research work of Nadya Ugryumova …”
Answer & Actions:
What is the meaning of the statement “The time courses of normalized intensity changes of the Raman bands at 967 cm–1 are in good agreement with the results of the OCT investigation.”?
In Fig. 12 an exponential (even if it is called “expotential£) fit is reported. The datasets seem to be too small for such a kind of fit.
The authors say that “The comparison of the results recorded with the OCT method and Raman spectroscopy, which are characterized by good timing, indicates the possibility of explaining changes in the optical parameters of the bones through changes in its internal structure – removal of the hydroxyapatite.”. This is not supported by data reported and related discussion.
Answer & Actions:
We gratefully acknowledge the Reviewer for these remarks. Additional explanations were given in the text.
“The character of changes was additionally approximated by the exponential function. There is a good time correlation between the decrease of the extinction coefficient of the bones (see Figure 6 and Figure 7) and the removal of the hydroxyapatite (see Figure 12).”
Reviewer 3 Report
The article The Optical Coherence Tomography and Raman spectroscopy for sensing of the bone demineralization process should be considered as a preliminary mention to the possible application of OCT in the field of osteology. The authors present the possibility of using OCT in human bone studies (using animal material).
Major remarks:
The analysis was carried out on only 22 (23) images (created in two experiments throughout the entire research cycle, that were considered separately). A small number of samples can generate significant errors by generating outlier values, without the possibility of correction by averaging the results from the replicates. It would be good to perform such tests on several types of bone materials, at least 2-3, so that any conclusions can be drawn. Overall, this research requires more experiments.
There is no conclusion in the abstract, what has been achieved in principle?
In the introduction, the authors provide few examples of the study of skeletal demineralization using visual methods. In the discussion, however, in general, there is no information if OCT can have advantages over existing methods? Can OCT be a complementary/independent test, e.g. in densitometry? What is the future of OCT in bone research? In Introduction/Discussion previous studies, even those based on dental bone material, are not discussed.
The Authors do not discuss also the OCT image structure in relation to bone tissue. For example, does the OCT image cover the periosteum and what is the OCT penetration depth in the properly mineralized and demineralized bone tissue?
Fig. 12 – This is of particular interest, but not discussed by the Authors, why the values from Raman spectroscopy (normalized intensity changes of bone tissue) from 3 to 24 h are stable, and they drop again after 24 h?
Figs. 6 and 7 are the same, so I can’t verify the results from the experiment.
Minor remarks:
The lines of the paragraphs should be numbered.
Line 4 (page 3) lacks a closing bracket.
All shortcuts should be clarified in the main text: IR, VIS-NIR.
The trend line in Fig 6 shouldn’t present an exponential fit?
Fig. 9. The value at point 24h is missed on the plot.
Discussion, line 12, page 13 – Figs 6 and 7 should be instead of 8 and 9.
Conclusions, last sentence – [37-40] instead of [3740].
Author Response
Thank you for your valuable comments. We have attached our answers below. We made changes to the manuscript based on them.
----------------------------------------------------------------------------------------
The article The Optical Coherence Tomography and Raman spectroscopy for sensing of the bone demineralization process should be considered as a preliminary mention to the possible application of OCT in the field of osteology. The authors present the possibility of using OCT in human bone studies (using animal material).
Major remarks:
The analysis was carried out on only 22 (23) images (created in two experiments throughout the entire research cycle, that were considered separately). A small number of samples can generate significant errors by generating outlier values, without the possibility of correction by averaging the results from the replicates. It would be good to perform such tests on several types of bone materials, at least 2-3, so that any conclusions can be drawn. Overall, this research requires more experiments.
Answer & Actions:
We gratefully acknowledge the Reviewer for these remarks. We added additional description of the aim of the research to the main body of the manuscript. This paper present preliminary stage of the research and our aim was to check whether we can obtain the reasonable quality signals and process them to get useful information. In further stages we are going to provide biostatistics and carry out calibration of sensing system based on larger set of the samples.
There is no conclusion in the abstract, what has been achieved in principle?
Answer & Actions:
We gratefully acknowledge the Reviewer for these remarks. We tried to show more precisely and clearly aim of the research and achievement to the abstract. We consider that getting reasonable quality of the signals and extracting data form them confirming demineralization of the bones as achievement of this stage of research.
In the introduction, the authors provide few examples of the study of skeletal demineralization using visual methods. In the discussion, however, in general, there is no information if OCT can have advantages over existing methods? Can OCT be a complementary/independent test, e.g. in densitometry? What is the future of OCT in bone research? In Introduction/Discussion previous studies, even those based on dental bone material, are not discussed.
Answer & Actions:
We gratefully acknowledge the Reviewer for these remarks. We added additional information to the Introduction.
“The purpose of this study is to determine whether bone demineralization can be estimated by means of complementary optical methods. As the bones are highly scat-tering media, the authors decided to use the optical coherence tomography (OCT, pio-neered by David Huang, James G. Fujimoto, and co-workers in 1991 [18]). Besides OCT, other measurement techniques for highly scattering media investigation, such as, e.g., methods using integration spheres [19,20], time-of-flight spectroscopy [21,22], photoa-coustic microscopy and spectroscopy [23,24], optical diffuse tomography [25,26], opti-cal diffraction tomography [27,28], optical projection tomography (or optical transmis-sion tomography) [29,30], transmission optical coherence tomography [31,32], are known. However, these methods either may only be performed in vitro (which is a major limitation if future clinical applications are considered) or cannot provide addi-tional information on bone properties (such as birefringence) with high spatial resolu-tion, unlike polarization-sensitive OCT (PS-OCT).
OCT is an optical measurement technique used for investigations of a wide range of scattering materials [33]. OCT enables surface and subsurface examination of dif-ferent types of materials to be performed in a non-contact and non-destructive way [34]. With the aid of OCT, one can analyze the depth structure of investigated materials with a measurement resolution of a few micrometers, high sensitivity, and a dynamic range [35]. Nowadays, the advantages of OCT make it applied in medical treatments, especially in ophthalmology, dermatology, stomatology, and endoscopy, and also in industry and science. The OCT measurements are based on selective, high-resolution detection of the light backscattered from the particular scattering points located inside the investigated object, which is needed for 2D and 3D tomography imaging. To per-form such measurements, the OCT uses low-coherence interferometry (LCI). The LCI, also known as white light interferometry (WLI), is an attractive measurement method offering high measurement resolution, high sensitivity, and measurement speed. Since both hydroxyapatite and the collagen fibers are birefringent [36,37], it has been decided to use PS-OCT, which enables simultaneous measurements of the scattering properties of the bones and their polarization properties. PS-OCT has already been used in investigations of hydroxyapatite-containing tis-sue specimens, namely teeth (tooth enamel is an anisotropic and weakly scattering medium that changes the state of polarization of backscattered optical radiation [12,38–41]), as well as collagen-containing tissues (e.g., skin [42,43], articular cartilage [44–47], meniscus [46], tendons [48], tumor [49,50], and atherosclerotic plaques [51] or coronary plaque [52] in blood vessels).”
The Authors do not discuss also the OCT image structure in relation to bone tissue. For example, does the OCT image cover the periosteum and what is the OCT penetration depth in the properly mineralized and demineralized bone tissue?
Answer & Actions:
We gratefully acknowledge the Reviewer for these remarks. We discussed the problems of limited depths of penetration of the signal.
Fig. 12 – This is of particular interest, but not discussed by the Authors, why the values from Raman spectroscopy (normalized intensity changes of bone tissue) from 3 to 24 h are stable, and they drop again after 24 h?
Answer & Actions:
We gratefully acknowledge the Reviewer for these remarks. It is partially consequence of using two samples. It is also the result of our concept of maximum, but independent stretching of both charts to better visualize possible trend lines. During the first 24 hours (which we wanted to show on separated chart), the process was the fastest, later it changed rather slowly.
Figs. 6 and 7 are the same, so I can’t verify the results from the experiment.
Answer & Actions:
We gratefully acknowledge the Reviewer for these remarks. It was obvious error. It has been corrected.
----------------------------------------------------------------------------------------
Minor remarks:
The lines of the paragraphs should be numbered.
Line 4 (page 3) lacks a closing bracket.
All shortcuts should be clarified in the main text: IR, VIS-NIR.
The trend line in Fig 6 shouldn’t present an exponential fit?
Fig. 9. The value at point 24h is missed on the plot.
Discussion, line 12, page 13 – Figs 6 and 7 should be instead of 8 and 9.
Conclusions, last sentence – [37-40] instead of [3740].
Answer & Actions:
We gratefully acknowledge the Reviewer for these remarks. The text was corrected.
Reviewer 4 Report
The authors have employed optical means using Raman and optical tomography for sensing bone demineralization. The manuscript read well and could be interest to readers in the field. However, there are a few issues that need to be addressed before publication.
- It would be good to provide a bit of background about Coherence Tomography and Raman spectroscopy in the introduction section and perhaps including relevant citation. Example of using Raman for biosensing doi.org/10.1016/j.snb.2020.128703
- The Raman section needs to be expanded. Having a control is crucial for Raman sensing and that is missing in the Figure.11.
Author Response
Thank you for your valuable comments. We have attached our answers below. We made changes to the manuscript based on them.
---------------------------------------------------------------------------------------
The authors have employed optical means using Raman and optical tomography for sensing bone demineralization. The manuscript read well and could be interest to readers in the field. However, there are a few issues that need to be addressed before publication.
- It would be good to provide a bit of background about Coherence Tomography and Raman spectroscopy in the introduction section and perhaps including relevant citation. Example of using Raman for biosensing doi.org/10.1016/j.snb.2020.128703
Answer & Actions:
We gratefully acknowledge the Reviewer for these remarks. Although our aim was to use as simple Raman system as possible for biosamples (we used fiber-optic system with 830 nm laser) we mentioned also more advanced Raman techniques in the Introduction
“The more complex Raman techniques can be also used for biomaterials investigation. It includes surface-enhanced Raman spectroscopy (SERS) which enables a significant in-crease of usually weak Raman signal by use of additional interaction of laser radiation with selected particles [56]. Also, Raman microscopy with dedicated sample selection can be used to remove signals from unwanted sources [57]. However, these techniques require special sample pre-treatment thus they were not introduced at this stage of the research as the aim of the research was to develop a minimally invasive sensing sys-tem.”
Moreover, we added two additional references.
- Keshavarz, M.; Rezaul Haque Chowdhury, A.K.M.; Kassanos, P.; Tan, B.; Venkatakrishnan, K. Self-assembled N-doped Q-dot carbon nanostructures as a SERS-active biosensor with selective therapeutic functionality. Sens. Actuators, B 2020, 323, 128703. https://doi.org/10.1016/j.snb.2020.128703
- Neugebauer, U.; Kurz, C.; Bocklitz, T.; Berger, T.; Velten, T.; Clement, J.H.; Krafft, C.; Popp J. Raman-Spectroscopy Based Cell Identification on a Microhole Array Chip. Micromachines 2014, 5, 204–215. https://doi.org/10.3390/mi5020204
- The Raman section needs to be expanded. Having a control is crucial for Raman sensing and that is missing in the Figure.11.
Answer & Actions:
We gratefully acknowledge the Reviewer for these remarks. The aim of the preliminary work presented in the project was to confirm bone demineralization using simple optical sensing systems, which in the case of Raman spectroscopy was achieved based on the disappearance of the 967 cm-1 spectral line. The disappearance of this line is visible in Figure 11. In the next stages of research, we intend to expand the biostatistics based on an increased number of measurements, which will also allow for better control of the remaining spectrum components and processed data. It should also be taken into account that we tested unprepared samples for Raman measurements and used a simple sensory system, so that raw data (Figure 11) may have undesirable background components that were then removed by data processing.
Round 2
Reviewer 1 Report
Most of my comments have been well addressed by the author. The only concern for me is the number of samples for each group. The author still haven't provide the exact number of samples in revised manuscript. Although PS-OCT images are composed of series of A-scans for each sample, it can not eliminate individual variability of the samples. Although the author planned to take into account the individual variability in next stage, the current difference between B24h and B14D in each measurement make the results look less convincing. Taken individually, both B24H and B14D are enough to support the hypothesis. Since this manuscript is a feasibility study, maybe the author can consider using one of them instead of put them together.
Author Response
Comment
Most of my comments have been well addressed by the author. The only concern for me is the number of samples for each group. The author still haven't provide the exact number of samples in revised manuscript. Although PS-OCT images are composed of series of A-scans for each sample, it can not eliminate individual variability of the samples. Although the author planned to take into account the individual variability in next stage, the current difference between B24h and B14D in each measurement make the results look less convincing. Taken individually, both B24H and B14D are enough to support the hypothesis. Since this manuscript is a feasibility study, maybe the author can consider using one of them instead of put them together.
Answer & Action
We would like to thank the Reviewer for the valuable comments. As we mentioned there were preliminary studies. Thus, we made two series of the measurements. One was dedicated for early stage of the process (24 hours with short time interval), while the second one - for whole process. We used two bones form the each serie: one for PS-OCT and one for Raman spectroscopy. We realize that it causes the results slightly less convincing, however we trust it would be clear for the reader if we more precisely describe the experiment and its limitations.
We added following explanation to the manuscript.
(2 pieces – one for PS-OCT and one for Raman spectroscopy)
Reviewer 2 Report
Not all the raised questions were fully solved. Especially, Fig.12 was not changed. In addition, the "trebd lines" reported in some figures can be misleading. Table 1 still seems not useful.
Author Response
Comment
Not all the raised questions were fully solved. Especially, Fig.12 was not changed. In addition, the "trebd lines" reported in some figures can be misleading. Table 1 still seems not useful.
Answer & Action
We would like to thank the Reviewer for the valuable comments. As we mentioned there were preliminary studies. Thus, we made two series of the measurements. One was dedicated for early stage of the process (24 hours with short time interval), while the second one - for whole process. We used two bones form the each serie: one for PS-OCT and one for Raman spectroscopy. We realize that it causes the results slightly less convincing, however we trust it would be clear for the reader if we more precisely describe the experiment and its limitations.
We added following explanation to the manuscript.
(2 pieces – one for PS-OCT and one for Raman spectroscopy)
Berceuse of limited amount of data it was quite difficult to significantly improve Figure 12. However, we trust is shows that in view of making two separate measurements for a different time period, it is better to show the results on separate graphs than to combine them, because they will not be more readable.
The use of exponential fit is of course an approximation, which we do not hide. It is to serve, like the whole of Figure 12, to indicate that the demineralization process in the conducted expert is most intense in the initial stage (first hours, first day) of the experiment, which will be confirmed by both methods.
Table 1 may seems not useful as it refers to different methods. However, according to us it has some advantages and we prefer to leave it in recent, improved form. The table provides a quick comparison of selected key parameters of the measurement systems (in particular, optical signal sources), which allows the reader to pay attention, for example, to the possibility of slight differences in the depth of penetration of the measurement signal.
Reviewer 3 Report
After the second review of the manuscript "The Optical Coherence Tomography and Raman spectroscopy 2 for sensing of the bone demineralization process," I would like to recommend this article to be published in Sensors.
The Authors improved most of the imperfections.
Author Response
We would like to thank the Reviewer for his work and prepared reviews with valuable comments for us.
Reviewer 4 Report
The authors have addressed all the comments and the manuscript is recommended for publication in its present form.
Author Response

(The authors gave the same response as above.)
